# The effect of acupuncture on oxidative stress: A systematic review and meta-analysis of animal models

Yu Zhao[1,2], Bo Zhou[2,3], Guangyin Zhang[2,3], Shixin Xu[2,4], Jipeng Yang[2,5], Shizhe Deng[2,5], Zengmin Yao[1,2], Qiang Geng[1,2], Bin Ouyang[1,2], Tian Xia[2,6]*

1 Andrology Department, First Teaching Hospital of Tianjin University of Traditional Chinese Medicine, Tianjin, China, 2 National Clinical Research Center for Chinese Medicine Acupuncture and Moxibustion, Tianjin, China, 3 Psychosomatic Medicine Department, First Teaching Hospital of Tianjin University of Traditional Chinese Medicine, Tianjin, China, 4 Tianjin Key Laboratory of Traditional Research of Traditional Chinese Medicine Prescription and Syndrome, First Teaching Hospital of Tianjin University of Traditional Chinese Medicine, Tianjin, China, 5 Acupuncture and Moxibustion Department, First Teaching Hospital of Tianjin University of Traditional Chinese Medicine, Tianjin, China, 6 Reproductive Center Department, First Teaching Hospital of Tianjin University of Traditional Chinese Medicine, Tianjin, China

* xiatian76@163.com

**Data Availability Statement:** All relevant data are within the paper and its Supporting Information files.

## Abstract

### Introduction

Oxidative stress is involved in the occurrence and development of multiple diseases. Acupuncture shows an excellent clinical efficacy in practical application but its mechanism remains unclear. This systematic review and meta-analysis was aimed at assessing the effect of acupuncture on oxidative stress in animal models.

### Methods

PubMed, Embase, and Web of Science database were retrieved for randomized controlled trials about acupuncture on oxidative stress in animal models from inception to August 2021. Two reviewers independently screened and extracted articles according to inclusion and exclusion criteria. We used the mean difference (MD)/standardized mean difference (SMD) to perform an effect size analysis and selected fixed-effect or random-effect models to pool the data, depending on a 95% confidence interval (CI).

### Results

A total of 12 studies comprising 125 samples were included in the quantitative meta-analysis. Compared with sham acupuncture, acupuncture (manual acupuncture, electropuncture, and laser acupuncture) reduced the level of malondialdehyde (SMD, −3.03; CI, −4.40, −1.65; $p < 0.00001$) and increased the levels of superoxide dismutase (SMD, 3.39; CI, 1.99, 4.79; $p < 0.00001$), glutathione peroxidase (SMD, 2.21; CI, 1.10, 3.32; $p < 0.00001$), and catalase (SMD, 2.80; CI, 0.57, 5.03; $p = 0.01$).

**Funding:** This research was funded by the Ministry of Science and Technology of the People's Republic of China (Project Grant no. 2018YFC1706001). There was no additional external funding received for this study. The funders had no role in study design, data collection and analysis, decision to publish, or preparation of the manuscript.

**Competing interests:** The authors have declared that no competing interests exist.

## Conclusion

This meta-analysis indicated that acupuncture can regulate oxidative stress by lowering the lipid peroxidation and activating the antioxidant enzyme system. In consideration of heterogeneity between studies, future studies should be performed by complying with strict standards and increasing sample size in animal experiments to reduce bias.

## Introduction

Oxidative stress is a classic biological process representing an imbalance between oxidative damage and oxidation resistance in vivo. Under normal physiological conditions, the generation of reactive oxygen species (ROS) in cells can not damage the biological function of the cells due to antioxidants released through cellular defense systems as a protective effect in vivo. A higher level of oxidative stress induced by various potential risk factors results in severe damage to all types of biomolecules when the levels of endogenous antioxidants are insufficient to quench the free radicals [1]. As a result, damage caused by oxidative stress can lead to the degeneration of proteins, lipids, and DNA/RNA, which in turn causes a series of pathological processes, including alteration of the genetic structure and DNA methylation, inhibition of cell proliferation and growth, the acceleration of cellular aging, and, ultimately cell death [2–4]. The modifications mentioned in the structure and function of cells can contribute, at least partially, to a variety of diseases including Alzheimer's disease (AD), cardiovascular disease (CD), ischemic stroke (IS), diabetes mellitus (DM), spinal cord injury(SCI), and male infertility (MI) [5–9]. For instance, on account of the characteristics of vulnerability to oxidative damage and a deficiency of antioxidants, the cells in the brain can be easily attacked by ROS [10–12], which then induce the oxidation of lipids, proteins, and DNA/RNA-also a common pathological feature in AD [13]-finally accelerating neuronal degeneration [14]. Endothelial dysfunction has been confirmed to play a vital role in the occurrence and development of CD [15]. The release of ROS mediated by nicotinamide adenine dinucleotide phosphate (NADPH) oxidases can impact the availability of a critical endothelium-derived relaxing factor [16], nitric oxide (NO) [17], which in turn leads to the repair dysfunction of vascular endothelial cells [15]. As an essential pathological factor, oxidative stress also accelerates neuronal cell death and apoptosis [18, 19] after sudden interruption or severe reduction of the blood flow and oxygen supply to the brain, causing local edemas and elevated intracranial pressure. This phenomenon further hinders the perfusion of the brain tissues and results in an IS [20–22]. Several studies suggest that oxidative stress plays a key role in triggering insulin resistance and the subsequent disruption of insulin signaling [23–25], and oxidative stress can also influence the development of secondary diabetic complications involving neuropathy, nephropathy, vascular disease, and retinopathy [25, 26]. Oxidative stress can also deteriorate SCI. Free radical can be produced and released after SCI, which causes cell death and tissue damage and subsequently aggravating SCI [27]. Moreover, an excessive production of ROS in sperm can impact the ability of mitochondria to acquire energy, causing sperm membrane and DNA damage and thereby leading to a reduction in the potential of the sperm to fertilize an egg and generate a healthy embryo [28–31]. In summary, oxidative stress can be as the cause of the pathology among multiple diseases and the contributor to disease progression [32]. It is generally clear that oxidative stress is involved in pathological development and could be the underlying etiology of multiple diseases. Hence, the key to improve or cure diseases may depend on the supplementation of antioxidants or the regulation of the balance in oxidative stress through other methods based on this specific mechanism.

Acupuncture, with a long history of being practiced for over 3000 years in China, has shown a clinical efficacy in treating several diseases worldwide [33]. In particular, acupuncture generated an excellent efficacy in treatment of the diseases outlined above [34–38]. Over the past few decades, a majority of studies regarding the therapeutic mechanisms of acupuncture in vivo have focused on neuroregulation, immunoregulation, metabolism, and gastrointestinal system [39–42]. An increasing number of studies have suggested that acupuncture generates a positive effect in regulation of the oxidative stress status in animal models [43–47]. To the best of our knowledge, no systematic meta-analysis has been published to analyze the effect and mechanism of acupuncture on oxidative stress in animal experiments to date. Therefore, we performed a systematic review and meta-analysis to investigate the experimental data that support the oxidation resistance of acupuncture with a particular focus on the related indicators.

## Materials and methods

This study was conducted by following the guidelines of the Preferred Reporting Items for Systematic Reviews and Meta-analyses (PRISMA) [48] and was registered in the PROSPERO database (registration number: CRD42021256081).

### Search strategy

Comprehensive article searching was undertaken by two authors independently in the PubMed, Embase, and Web of Science databases from inception to August 2021 with no limitation on publication language. To identify any additional relevant articles, the two observers manually reviewed the lists of references in the selected articles. There were no limits on the publication data.

The whole search strategies (mesh terms and all fields) in PubMed were as follows: (Animal Model OR Animal Models OR Laboratory Animal Models OR Laboratory Animal Model OR Experimental Animal Models OR Animal OR Animal Models, Experimental OR Experimental Animal Model) AND (Acupuncture) AND (Oxidative Stresses OR Antioxidative Stress OR Antioxidative Stresses OR Anti-oxidative Stress OR Anti oxidative Stress OR Anti-oxidative Stresses OR Oxidative Damage OR Oxidative Damages OR Oxidative Stress Injury OR Oxidative Stress Injuries OR Oxidative Injury OR Oxidative Injuries OR Oxidative Cleavage OR Oxidative Cleavages OR Oxidative DNA Damage OR Oxidative DNA Damages OR DNA Oxidative Damage OR DNA Oxidative Damages OR Oxidative and Nitrative Stress OR Oxidative Nitrative Stress OR Oxidative Nitrative Stresses OR Nitro-Oxidative Stress OR Nitro Oxidative Stress OR Nitro-Oxidative Stresses) AND (Sham acupuncture).

In Embase, the search string was (animal model:ab,ti OR animal disease model:ab,ti OR animal models:ab,ti) AND (acupuncture:ab,ti OR acupuncture therapy:ab,ti OR shonishin:ab, ti) AND (oxidative stress:ab,ti OR oxidant stress:ab,ti OR oxidant stresses:ab,ti OR oxidative stresses:ab,ti) AND (sham acupuncture:ab,ti).

### Inclusion and exclusion criteria

The inclusion criteria applied to the study selection were as follows: (1) animal models with diseases caused by oxidative stress; (2) any type of acupuncture treatment with explicit instructions for acupoint selection, intensity, duration of treatment, and period (manual acupuncture: steel needles inserted into specific acupoints based on the meridian and collateral theory in the form of intermittent rotation; electropuncture: implemented combined with electrical stimulation on needles, in particular the strength of the electric current or voltage; laser acupuncture: operated by focusing irradiation at specific points with a low intensity laser); (3) comparisons

with a control group that received a sham acupuncture intervention; and (4) any species, sex, weight, or age.

The exclusion criteria were as follows: (1) animal experiments in vitro or ex vivo, and studies in humans or silicon models; (2) combinations with other interventions (traditional Chinese medicine decoction, moxibustion, Chinese patent medicine, etc.); (3) case reports, literature reviews, and conference abstracts; and (4) full texts of studies not available.

## Study selection

After removing the duplicates, two reviewers screened the titles and abstracts to select the related studies to be imported into EndNote X7. Full text screening was then applied to identify the unique articles meeting the inclusion criteria. If there was a disagreement between the reviewers, it was resolved by consulting a third researcher through rigorous discussions.

## Data extraction

Information regarding each included study (e.g., authors, publication year, species, weight, acupoint selection, intervention, frequency or intensity, outcome measures, and treatment duration) was extracted by two reviewers independently. If the data were presented in the form of a graph, GetData Graph Digitizer (http://getdata-graph-digitizer.com/)(2021.9.30) was used to extract the numerical data from the diagrams [49].

## Risk of bias assessment

The SYRCLE RoB tool [50] was used independently by the two reviewers to evaluate the risk of bias (RoB). The tool contains 10 items involved in six aspects of bias (selection bias, performance bias, detection bias, attrition bias, reporting bias, and other biases). Scores of 'yes', 'no', and 'unsure' separately indicate a 'low', 'high', and 'unclear' RoB, respectively, and were shown on the Cochrane RoB tool [51].

## Data analysis

The experimental group (manual acupuncture, electropuncture, and laser acupuncture) and control group (sham acupuncture) data from the included studies were extracted and imported into Revman 5.3 software. When the outcome measures of all the included studies were on the same scale, the mean difference (MD) was used to perform the effect size analysis. Otherwise, the standardized mean difference (SMD) was used. Confidence intervals (CIs) of 95% were calculated for the effects of acupuncture on oxidative stress. The heterogeneity among the studies was classified according to the $I^2$ test. When the $I^2$ was $\leq$50% (low heterogeneity), a fixed-effect model was used. When the $I^2$ was >50% (high heterogeneity), a random-effect model was used. When the subgroups comprised at least two independent comparisons, subgroup analyses were performed. A sensitivity analysis was conducted to account for the risk of bias through a leave-one-out method operated in OpenMeta (Analyst) software, represented by a leave-one-out forest plot.

## Publication bias

We implemented the assessment of publication bias using a visual inspection of the funnel plot asymmetry and Egger's test of asymmetry [52]. If there were fewer than 10 studies associated with one outcome, the power of the assessment was too low to be performed according to the Cochrane recommendations. Egger's test of asymmetry was also invalid on the condition that the number of included studies was fewer than 20.

# Results

## Study selection

After a comprehensive search for articles in the databases, 40 articles were initially identified according to their titles and abstracts. Following a full text screening, several were eliminated based on the following reasons: duplicate publication (n = 3), publication language in Chinese (n = 5), not providing an intervention of any type of acupuncture (n = 5), not focusing on oxidative stress (n = 5), no related outcome measure provided (n = 1), no sham acupuncture as the control (n = 7), combined with another intervention (oral drugs) (n = 1), and full text unavailable (n = 1). Ultimately, a total of 12 studies comprising 125 animal models were included in the quantitative meta-analysis. The flow of searching the databases is displayed in Fig 1.

## Study characteristics

The characteristics of the 12 included studies are listed in Table 1. Of these studies, the animal models were all rats or mice, but of different breeds and ages. The number of samples per

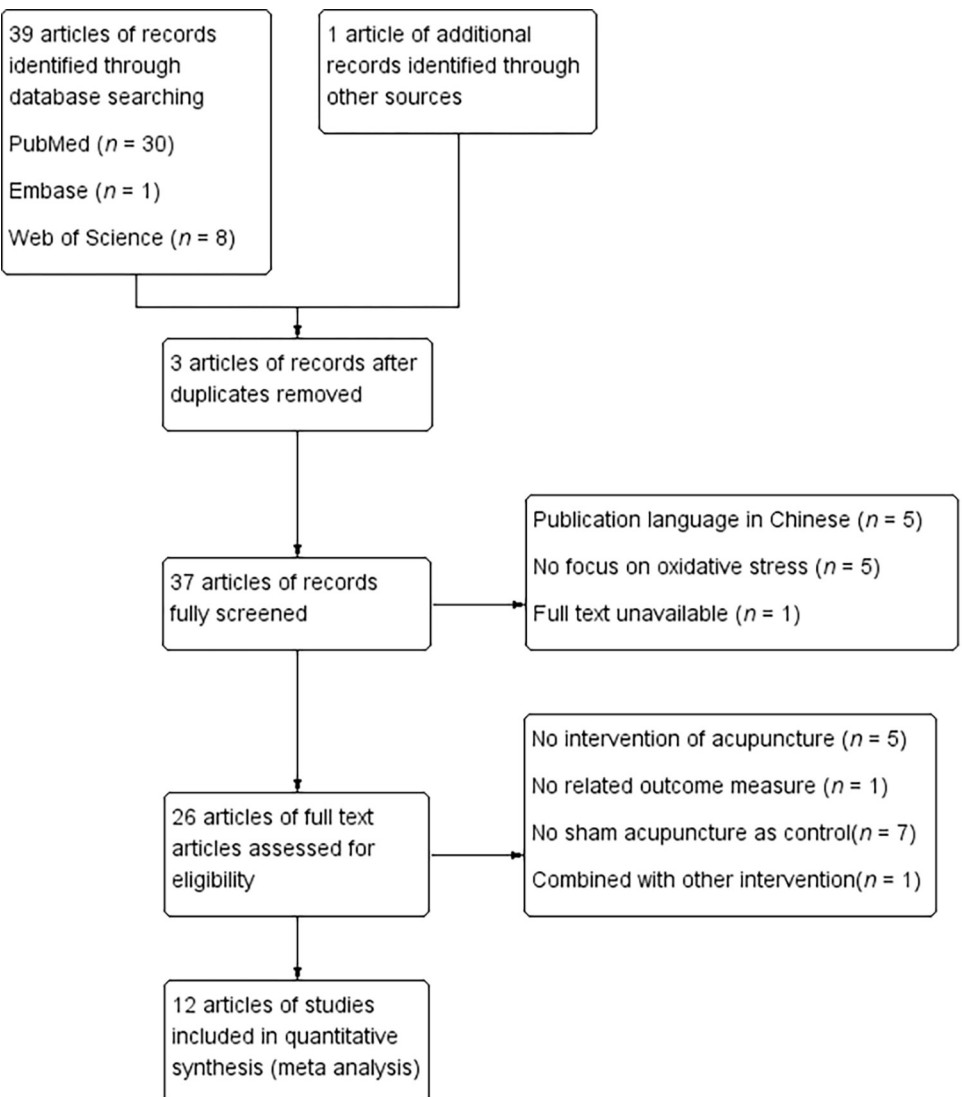

**Fig 1. Flow diagram of the systematic review and article search results.**

**Table 1. Characteristics of the included studies.**

| Author (Year) | Species (Sex) | Weight (g) | Num | Exp | Con | Sample | Acupoint Selection | Frequency/ Intensity | Indicators | Duration (Day) |
|---|---|---|---|---|---|---|---|---|---|---|
| Alvarado-Sanchez et al. [53] (2019) | Long Evans rats (female) | 250–300 | 14 | EA | SA | Spinal cord | GV4 | 2Hz/100 Hz / 5.2 mA | MDA / H2O2 / TBARS | ? |
| Siu et al. [54] (2005) | Sprague–Dawley rats (male) | 330–350 | 6 | EA | SA | Homogenate of brain tissue | GB20 / ST36 | 2 Hz / 0.7 V | TR / Trx / NADPH | 14 |
| Li et al. [55] (2020) | Sprague–Dawley rats (male) | 250–300 | 20 | EA | SA | Hippocampal tissues | LI11, ST36 / DU20 | 2Hz/15Hz / 1.5 mA | MDA / SOD | 10 |
| Leung et al. [56] (2016) | SHRs (male) | ? | 8 | EA | SA | Plasma | ST36 / LR3 | 2 Hz / 2 mA | NADPH | 30 |
| Tian et al. [57] (2018) | C57BL/6 wild-type mice (male) | ? | 12 | EA | SA | Plasma | ST36 | 10 Hz / 1–3 mA | MDA | 32 |
| Chang et al. [58] (2019) | Senescence-resistant mouse strain 8 (male) | ? | 10 | MA | SA | Hippocampal tissues | CV17 / CV12 / CV6 / SP10 / ST36 | ? | SOD / GSH-Px | 14 |
| Liu et al. [59] (2006) | Wistar rats (male) | 340 ± 40 | 9 | MA | SA | Hippocampal tissues | CV17 / CV12 / CV6 / ST36 / SP10 | Twisted at the speed of twice a second for 30 s | SOD / CAT / GSH-Px | 14 |
| Phunchago et al. [60] (2014) | Wistar rats (male) | 180–220 | 6 | MA | SA | Homogenate of brain tissue | HT7 | Twisted at the speed of twice a second for 60 s | MDA / SOD / GSH-Px / CAT | 14 |
| Fei-yi Z et al. [61] (2021) | Sprague–Dawley rats (male) | 200 ± 20 | 14 | MA | SA | Prefrontal cortex | GV20 / HT7 / SP6 / GV29 | ? | MDA / SOD / GSH-Px | 18 |
| Sutalangka et al. [62] (2013) | Wistar rats (male) | 180–220 | 6 | LA | SA | Hippocampal tissues | HT7 | 405 nm / 100 mW | MDA / SOD / GSH-Px / CAT | 14 |
| Jittiwat [63] (2017) | Wistar rats (male) | 300–350 | 10 | LA | SA | Cerebral cortex | GV20 | 810 nm / 100 mW | MDA / SOD / GSH-Px / CAT | 14 |
| Jittiwat [64] (2019) | Wistar rats (male) | 300–350 | 10 | LA | SA | Hippocampal tissues | GV20 | 810 nm / 100 µm | SOD / GSH-Px | 14 |

?: not mentioned; SHRs: spontaneously hypertensive rats; EA: electroacupuncture; LA: laser acupuncture; MA: manual acupuncture; SA: sham acupuncture; MDA: malondialdehyde; TBARS: thiobarbituric acid reaction substance; GSH-Px: glutathione peroxidase; SOD: superoxide dismutase; GSH: glutathione; CAT: catalase; TR: thioredoxin reductase; Trx: thioredoxin; NADPH: nicotinamide adenine dinucleotide phosphate.

group ranged from 6 to 20. Five studies experimented with electroacupuncture (EA) [53–57], four with manual acupuncture (MA) [58–61], and three with laser acupuncture (LA) [62–64]. Five studies sampled the hippocampal tissues from the rats for detection [55, 58, 59, 64], two used plasma [56, 57], two used a homogenate of the brain tissues [54, 61], one used the spinal cord [53], one used the prefrontal cortex [61], and one used the cerebral cortex [63].

### Risk-of-bias assessment

The risk-of-bias assessment of the included studies is shown in Fig 2a and the individual scores for the 10 items of each study are presented in Fig 2b. In total, all 12 studies described a random allocation but did not provide specific random methods, which resulted in all being classified "unclear". Only two studies [53, 59] did not describe the feeding conditions to ensure comparability of the baseline characteristics between the two groups. The risk of random housing was high in two studies [53, 54]. Across the studies, insufficient information led to an uncertainty of the risk of bias regarding the blinding of caregivers as well as the randomness and blinding of the outcome assessment. All studies recorded a complete outcome simultaneously without bias from other sources.

### Data extraction

Only two studies [58, 61] provided detailed data that were represented numerically. Other studies presented the experimental data graphically; therefore, GetData Graph Digitizer was used to obtain the numerical data.

### Malondialdehyde (MDA)

Seven of the 12 included studies measured the malondialdehyde (MDA) level and these data are shown in Fig 3. Given the high heterogeneity among the included studies, a random-effect model was used to pool the data. Compared with sham acupuncture, acupuncture significantly decreased the level of MDA (SMD, -3.03; CI, -4.40, -1.65; $p < 0.00001$). A sensitivity analysis was performed and illustrated that the effect sizes were stable; the elimination of a single study did not impact the significance.

### Superoxide Dismutase (SOD)

Eight of the included studies measured the superoxide dismutase (SOD) level and the pooled data of these are presented in Fig 4. Similarly, a random-effect model was performed due to a high level of heterogeneity between the individual studies. In the meta-analysis, acupuncture was associated with a significant improvement on the SOD level (SMD, 3.39; CI, 1.99, 4.79; $p < 0.00001$). A sensitivity analysis was performed and illustrated that the effect sizes were stable; the elimination of a single study did not impact the significance.

### Glutathione Peroxidase (GSH-Px)

Seven of the included studies measured the glutathione peroxidase (GSH-Px) level and the pooled data of these seven studies are displayed in Fig 5. Acupuncture increased the level of GSH-Px (SMD, 2.21; CI, 1.10, 3.32; $p < 0.00001$). A sensitivity analysis was performed and illustrated that the effect sizes were stable; the elimination of a single study did not impact the significance.

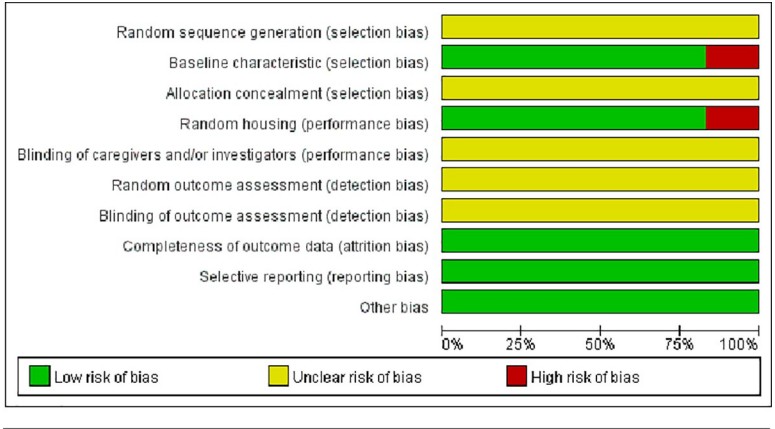

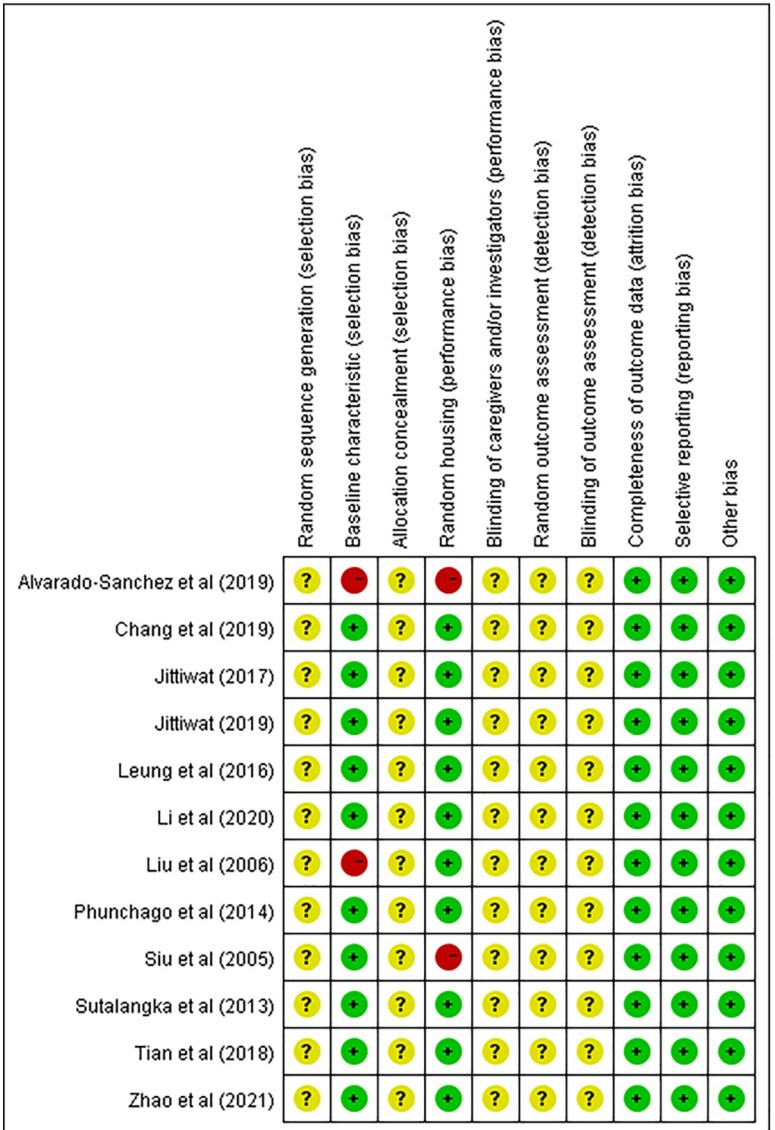

**Fig 2. Risk of bias.** (a) Following the SYRCLE tool, each risk-of-bias item is displayed as a percentage according to all included studies. (b) Individual risk of bias of the 10 items in the SYRCLE tool on all included studies, representing 'yes', 'no', or 'unclear'.

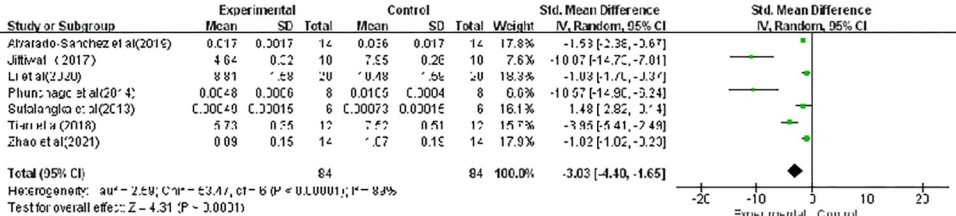

**Fig 3. Forest plot showing the effect of acupuncture on MDA levels.** CI: confidence interval; IV: inverse variance; MD: mean difference; SD: standard deviation; WMD: weighted mean difference.

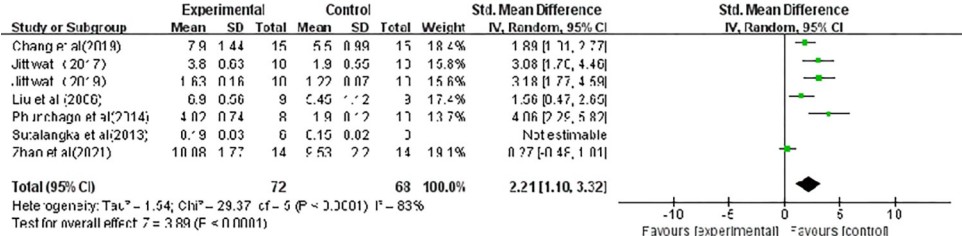

**Fig 4. Forest plot showing the effect of acupuncture on SOD levels.** Note: CI: confidence interval; IV: inverse variance; MD: mean difference; SD: standard deviation; WMD: weighted mean difference.

**Fig 5. Forest plot showing the effect of acupuncture on GSH-Px levels.** Note: CI: confidence interval; IV: inverse variance; MD: mean difference; SD: standard deviation; WMD: weighted mean difference.

## Catalase (CAT)

Four of the included studies measured the catalase (CAT) level and the pooled data of these are displayed in Fig 6. Acupuncture increased the level of CAT (SMD, 2.80; CI, 0.57, 5.03; p = 0.01). A sensitivity analysis was performed and illustrated that the effect sizes was stable; the elimination of a single study did not impact the significance.

## Subgroup analysis

Subgroup analyses were performed based on the type of intervention and the species used in the experimental animals. Four studies [58–61] that used manual acupuncture, two studies [55, 61] that used Sprague–Dawley rats and evaluated MDA, and three studies [62–64] that used laser acupuncture and evaluated GSH-Px showed a low heterogeneity between each other. The pooled data of the subgroup analyses are shown in Table 2.

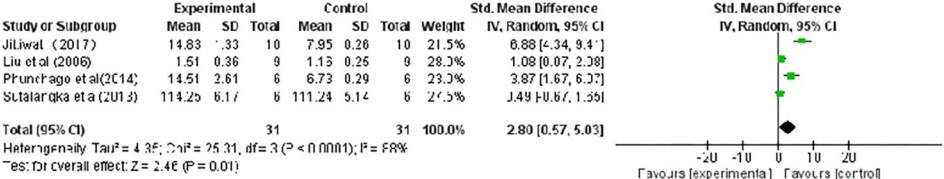

**Fig 6. Forest plot showing the effect of acupuncture on CAT levels.** Note: CI: confidence interval; IV: inverse variance; MD: mean difference; SD: standard deviation; WMD: weighted mean difference.

## Discussion

In this study, we demonstrated that acupuncture can regulate oxidative stress in animal models in different organs including the brain, vessels, stomach, and spinal nerves compared with sham acupuncture. Twelve studies were eligible for the meta-analysis, which showed that acupuncture could significantly reduce the MDA level and increase the SOD, GSH-Px, and CAT levels. A high level of lipid peroxidation can overwhelm antioxidant defent system in vivo and induce cell apoptosis or other pathological reaction [65]. This process will elevate the concentration of MDA, which can reflect the increase of free radical production and the degree of oxidative stress [66]; SOD, GSH-Px and CAT are all excellent antioxidants in vivo and participation of antioxidant systems and play a part through eliminating oxygen free radicals [67]. Hence, we confirmed that acupuncture stimulating specific acupoints decreased the levels of lipid peroxidation and activated the inherent antioxidant enzyme system to balance the oxidative stress status in several tissues and organs. A data analysis of the relevant indicators demonstrated that acupuncture plays an important role in regulating the oxidative stress reaction; furthermore, it provides an excellent curative effect through multiple target points.

From a clinical point of view, independent of the type of intervention applied (MA, EA, or LA), all studies provided stimulation at a specific point and produced an effect on the tissue. We pooled the data from the included studies using different types of intervention. Based on the subgroup analysis, we observed that the results of LA on MDA (SMD, -5.99; CI, -15.20, 3.21; $p = 0.20$), and MA (SMD, 2.29; CI, -0.42, 5.01; $p = 0.10$) and LA (SMD, 3.58; CI, -2.68, 9.84; $p = 0.26$) on CAT had no statistical differences. Apart from the subgroup analysis on

**Table 2. Subgroup analyses of studies using different types of acupuncture and species of experimental animals.**

| Indicator | Intervention/Species | SMD (95% CI) | I2 | p (Heterogeneity) |
|---|---|---|---|---|
| MDA | Electropuncture | −2.02 (−3.38, −0.65) | 84% | 0.002 |
| | Laser acupuncture | −5.99 (−15.20, 3.21) | 95% | < 0.00001 |
| | Wistar rats | −7.44 (−14.71, −0.18) | 94% | < 0.00001 |
| SOD | Manual acupuncture | 1.51 (0.82, 2.21) | 38% | 0.19 |
| | Laser acupuncture | 9.70 (5.12, 14.28) | 77% | 0.01 |
| | Wistar rats | 6.30 (2.81, 9.78) | 91% | < 0.00001 |
| | Sprague–Dawley rats | 1.61 (1.06, 2.17) | 0 | 0.75 |
| GSH-Px | Manual acupuncture | 1.78 (0.48, 3.07) | 84% | 0.0003 |
| | Laser acupuncture | 2.55 (1.44, 3.67) | 49% | 0.14 |
| | Wistar rats | 2.56 (1.60, 3.51) | 59% | 0.05 |
| CAT | Manual acupuncture | 2.29 (−0.42, 5.01) | 80% | 0.02 |
| | Laser acupuncture | 3.58 (−2.68, 9.84) | 95% | < 0.00001 |

Note: MDA: malondialdehyde; SOD: superoxide dismutase; GSH-Px: glutathione peroxidase; CAT: catalase.

SOD, all subgroups overlapped on the confidence interval, which suggest that there were interactions between the variables. Therefore, the results of the subgroup analysis did not affect the results of the comprehensive analysis.

Despite applying an experimental design and being highly controlled, there was still considerable heterogeneity among the studies included in this article. The subgroup analysis based on the different types of intervention and animal model indicated that heterogeneity still existed between the studies. This phenomenon may be due to differences in the forms of animal rearing, experiment reagents, sampling, and acupuncture prescriptions. As a characteristic of the acupoint selection in acupuncture, a systematic and standardized treatment protocol may not have been implemented. Therefore, a greater uniformity of the protocol design and of the animal models would have minimized the heterogeneity between the studies, enhanced the comparability of results, reduced experimental errors, and increased the grade of evidence. However, because of the small sample size, this subgroup analysis lacked statistical power.

Over the years, the therapeutic mechanism of acupuncture has remained unclear and, as a result, has always been a research focus [68]. The reason for this phenomenon might be that the practical application of this ancient technology is based on the meridian and collateral theory, which is beyond understanding and unobservable in the human anatomy [69]. Considering its beneficial effects, it is necessary to explore and establish the potential mechanisms of acupuncture in treating diseases. With cumulative animal studies being performed [43, 70–72], the relationship between acupuncture and oxidative stress has become clear. Despite a high heterogeneity between the included studies, the trend of improvement of oxidative stress by acupuncture was displayed through the pooled data, which were consistent with previous studies [73, 74]. Resisting oxidative stress is known to involve several aspects [75], such as (i) the inhibition of the production of ROS; (ii) the elimination of ROS by antioxidant enzymes or another signal pathway; and (iii) the repairing of proteins, lipids, or DNA attacked by ROS. A study indicated that EA stimulation at GV20 in diabetic rats with a cerebral ischemia could inhibit the activation of NOX, a major ROS-producing enzyme, and lower the MDA content and ROS formation [76]. Another study reported that manual acupuncture at Tanzhong (CV17), Zhongwan (CV12), Qihai (CV6), Sanyinjiao (ST36), and Xuehai (SP10) promoted the activities of the total SOD and decreased the level of MDA in mitochondria [77]. EA stimulation at GV20 and ST36 attenuated oxidative stress via increased CAT and SOD activity in the serum and hippocampus [78], which suggested that acupuncture could regulate oxidative stress through antioxidant enzymes in vivo. Meanwhile, EA stimulation at ST36 and SP6 can lower the total SOD activity and inhibit the $H_2O_2$ and MDA level in corpus striatum[79]. At a molecular level, acupuncture similarly has been found to repair proteins, lipids, or DNA attacked by ROS [44].

As discussed, exogenous supplementation of antioxidants plays an essential role in clinical practice. The application of various antioxidants has been proven with excellent clinical effects [80, 81]. Compared with exogenous antioxidants that need to be administered orally, acupuncture has several advantages such as it not being metabolized in vivo as well as economy and acceptability. Although this meta-analysis focused on animal studies, the data from this study support the function of acupuncture on oxidative stress and point to a direction for future clinical studies as a basis or guidance for human disease.

To our knowledge, this is the first systematic review of the effects of acupuncture on oxidative stress in animal studies. A comprehensive search was applied in multiple databases for the full texts of all identified articles. The SYRCLE RoB tool was used to assess the quality of the studies and data related to oxidative stress were extracted. Subgroup and sensitivity analyses were performed to validate our discoveries.

There are a few limitations to this study. First, the published language was limited to English. Second, one study was excluded as the full text was unavailable. Finally, the quality assessment using the SYRCE RoB tool reflected that the included studies did not provide sufficient information to reduce the risk of performance and detection bias.

## Conclusion

In conclusion, we observed that acupuncture significantly decreases the level of MDA and increases the levels of SOD, GSH-Px, and CAT. The data from existing experimental studies suggest that acupuncture can regulate oxidative stress status among multiple organs and tissues in animal models. However, more studies, especially clinical studies, are still needed to further explore and justify the oxidation resistance of acupuncture. All animal studies had major methodological limitations including a small sample size, performance bias, and detection bias. Therefore, future studies should be performed according to strict standards to reduce bias and by increasing the sample size in animal experiments.

## Supporting information

**S1 File. PRISMA_2020_checklist.**
(DOCX)

**S1 Fig. (SOD) Leave-one-out_Forest_Plot.**
(TIF)

**S2 Fig. (MDA) Leave-one-out_Forest_Plot.**
(TIF)

**S3 Fig. (CAT) Leave-one-out_Forest_Plot.**
(TIF)

**S4 Fig. (GSH-Px) Leave-one-out_Forest_Plot.**
(TIF)

## Acknowledgments

The authors thank Ruihong Ma for assistance with reference management and the creation of tables and figures for this manuscript.

## Author Contributions

**Conceptualization:** Yu Zhao.

**Formal analysis:** Jipeng Yang, Shizhe Deng.

**Funding acquisition:** Shizhe Deng.

**Investigation:** Bo Zhou, Guangyin Zhang, Shixin Xu.

**Methodology:** Bo Zhou, Guangyin Zhang, Shixin Xu.

**Supervision:** Tian Xia.

**Writing – original draft:** Yu Zhao.

**Writing – review & editing:** Guangyin Zhang, Shixin Xu, Zengmin Yao, Qiang Geng, Bin Ouyang, Tian Xia.

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
