## [Decision Letter · Decision Letter 0]

17 Apr 2022

PONE-D-22-01762The Effect of Acupuncture on Oxidative Stress: A Systematic Review and Meta-Analysis of Animal ModelsPLOS ONE

Dear Dr. Xia,

Thank you for submitting your manuscript to PLOS ONE. After careful consideration, we feel that it has merit but does not fully meet PLOS ONE’s publication criteria as it currently stands. Therefore, we invite you to submit a revised version of the manuscript that addresses the points raised during the review process.

We look forward to receiving your revised manuscript.

Kind regards,

Ghulam Md Ashraf, Ph.D.

Academic Editor

PLOS ONE

Journal Requirements:

"All relevant data are within the manuscript and its Supporting Information files."

3. We note that this manuscript is a systematic review or meta-analysis; our author guidelines therefore require that you use PRISMA guidance to help improve reporting quality of this type of study. Please upload copies of the completed PRISMA checklist as Supporting Information with a file name “PRISMA checklist

Reviewers' comments:

Reviewer's Responses to Questions

**Comments to the Author**

1. Is the manuscript technically sound, and do the data support the conclusions?

Reviewer #1: Yes

Reviewer #2: Partly

Reviewer #3: Partly

2. Has the statistical analysis been performed appropriately and rigorously? 

Reviewer #1: Yes

Reviewer #2: I Don't Know

Reviewer #3: I Don't Know

3. Have the authors made all data underlying the findings in their manuscript fully available?

Reviewer #1: Yes

Reviewer #2: Yes

Reviewer #3: Yes

4. Is the manuscript presented in an intelligible fashion and written in standard English?

Reviewer #1: No

Reviewer #2: Yes

Reviewer #3: Yes

5. Review Comments to the Author

Reviewer #1: Thank you for writing this amazing research, however, the research needs minor revision as indicated below:

1. The introduction is well written however, there is need to justify and explain deeply about the literatures reported.

3. The Methodology is well written if possible make reference to the part where needed.

3. The results is well explained, however, the discussion need support of more scholarly literatures to establish your viewpoint.

4. The conclusion should reflect the limitations and strength of the research.

Reviewer #2: This study mined literature to show that acupuncture can regulate oxidative

stress in animal models. 12 studies were included in the meta-analysis, that showed that acupuncture reduces the MDA level and increase the SOD, GSH-Px, and CAT

levels. This meta-analysis indicates that acupuncture increases levels of ROS scavengers like CAT, SOD and GSH-Px. The study provides valuable insights, however, I have the following concerns;

Acupuncture activates lowering lipid peroxidation and antioxidant enzyme system: comment on mechanism in the light of available literature.

Sample size seems to be small. The analysis could have been extended to the non-English studies.

What was the rationale for using sham acupuncture as control?

The acupoints are not explicitly mentioned in the text unless I missed it.

Table 1 needs to be formatted. Due to formatting problems, it was really hard to understand table 1.

What is the ‘other bias’ in figure 2?

The legends of tables/figs should describe in sufficient details the about the data. I feel legends are not descriptive at all and makes hard to make conclusions.

I think many claims in the discussion need to be toned down

Lines 320-322 not sure if just increase of CAT, SOD and GST-Px would mean ‘excellent oxidation resistance’

Lines 399-401 How did you confirm “that acupuncture reduces the production of ROS and

activates the antioxidant system in vivo” and how is it “evidence to prove the mechanism of acupuncture in treating multiple diseases”. This is merely a meta-analysis.

Why is not the supplementary data cited in the text? How does one review it then.

Reviewer #3: The authors conducted a systematic review and meta-analysis of the published article assessing the effect of acupuncture on oxidative stress in animal models. The topic is relevant in the context of alternative medicine and may be of interest to the readers. However, following points need to be addressed -

1. Literature search was undertaken from “inception to August 2021” (line# 28 and 117). What does inception mean?

2. The effect of acupuncture on other disease areas could have been discussed briefly in the introduction section as supporting information to discuss the relevance of this study.

3. Please explain/elaborate what indicators (MDA, SOD, GSH-Px, CAT) are and discuss briefly how they are related to oxidative stress.

4. In line# 15, what does “full text screening was then applied” mean?

5. The font used in the manuscript is not reading friendly and the format of table 2 made it hard to follow.

6. Last line of conclusion (line# 405-408) seems irrelevant and does not fit to the context.

7. Correction: “increased” in line# 36. Small letter in “criteria” in line# 29.

6. PLOS authors have the option to publish the peer review history of their article (what does this mean?). If published, this will include your full peer review and any attached files.

Reviewer #1: **Yes: **Abdullahi Tunde Aborode

Reviewer #2: No

Reviewer #3: No

---

## [Author Response · Author response to Decision Letter 0]

28 May 2022

Resopnd to editor:

#1 Thanks for your comments. We have modified the templates of our manuscript to meet PLOS ONE's style requirements.

#2 Thanks for your comments. We have provided fund statements as required.

#3 Thanks for your comments. The PRISMA checklist has been uploaded in the system.

Respond to reviewers:

Reviewer #1: Thank you for writing this amazing research, however, the research needs minor revision as indicated below:

1# The introduction is well written however, there is need to justify and explain deeply about the literatures reported.

Reply: Thanks for your considerable and valuable advice. We have added related content to justify and explain about the literatures reported, which is shown in line 88-90 in the new version of manuscript.

2# The Methodology is well written if possible make reference to the part where needed.

Reply: Thanks for your considerable and valuable advice.We have added relevant reference to support relevant content, which is shown in line 163(reference number 49) in the new version of manuscript.

3#The results is well explained, however, the discussion need support of more scholarly literatures to establish your viewpoint.

Reply: Thanks for your considerable advice. The scholarly literatures to support our view point have been supplemented, which is shown in line 354-356 in the new version of manuscript.

4#The conclusion should reflect the limitations and strength of the research.

Reply: Thanks for your considerable advice. The conclusion included the statement of the limitations and strength of our study in the old version, which is shown in line 366-375 in the new version of manuscript.

Reviewer #2

Reviewer #2: This study mined literature to show that acupuncture can regulate oxidative stress in animal models. 12 studies were included in the meta-analysis, that showed that acupuncture reduces the MDA level and increase the SOD, GSH-Px, and CAT levels. This meta-analysis indicates that acupuncture increases levels of ROS scavengers like CAT, SOD and GSH-Px. The study provides valuable insights, however, I have the following concerns;

Acupuncture activates lowering lipid peroxidation and antioxidant enzyme system: comment on mechanism in the light of available literature.

1# Sample size seems to be small. The analysis could have been extended to the non-English studies.

Reply: Thanks for your valuable comments. We performed the literature search according to our search trategy, and finally obtained 40 articles. Among the including studies, all of them provided a small sample size. The reason for this may be due to the characteristics of animal studies based on our standpoint. It is very kind of you to point out the linguistic limitation in our retrieval strategy. We have revised our search strategy according to your comments and the new search results remained unchanged(five studies published in Chinese but not meet with our inclusion criteria). The related content is shown in line 115 in the new version of manuscript.

2# What was the rationale for using sham acupuncture as control?

Reply: Thanks for your valuable comments. As known, it is critical to perform blinding method in the implementation process of randomized controlled trials. Blinding method can lower the bias caused by various factors and improve the authenticity and reliability of research results, as being the same to animal studies. In conclusion, this is our rationale to select sham acupuncture as control.

3# The acupoints are not explicitly mentioned in the text unless I missed it.

Reply: Thanks for your comments. We have shown the selected acupoints from each including study in the Table 1(Characteristics of the included studies).

4# Table 1 needs to be formatted. Due to formatting problems, it was really hard to understand table 1. What is the ‘other bias’ in figure 2?

Reply: Thanks for your valuable comments. We have adjusted the format of Table 1 to make it easier to understand. 

The ‘other bias ’ was: except the five biases above, there may be other factors caused bias in the including studies. For instance, incomplete information to estimate, bias associated with the study design used, early termination in studies, clear imbalance in baseline, claims of deception and other issues.

5# The legends of tables/figs should describe in sufficient details the about the data. I feel legends are not descriptive at all and makes hard to make conclusions.

Reply: We politely disagree. Except supplementary files, there are two tables and six figures in this meta-analysis . Table 1 shows the characteristics of the included studies in detail. Table 2 shows the results of subgroup analysis(displayed as SMD, I2 and P). Both of them all display the detail data needed to express or state. Figure 1 show the article research process. Figure 2 (a and b) show the bias of included studies. Figure 3-6 called forest plot keep detailed records of data from various studies and pooled data(95%CI and heterogeneity). From our point of view, all the figures can describe the details about the data. We have searched several similar published meta-analysis and finally found that they all illustrate the data in this way.

Related reference: 

1.Bei T, Yang L, Huang Q, Wu J, Liu J. Effectiveness of bone substitute materials in opening wedge high tibial osteotomy: a systematic review and meta-analysis. Ann Med. 2022;54(1):565-577. doi:10.1080/07853890.2022.2036805

2. Sherafati N, Bideshki MV, Behzadi M, Mobarak S, Asadi M, Sadeghi O. Effect of supplementation with Chlorella vulgaris on lipid profile in adults: A systematic review and dose-response meta-analysis of randomized controlled trials. Complement Ther Med. 2022, 66:102822. doi: 10.1016/j.ctim.2022.102822. 

3. Grossi U, Gallo G, Ortenzi M, Piccino M, Salimian N, Guerrieri M, Sammarco G, Felice C, Santoro GA, Di Saverio S, Di Tanna GL, Zanus G. Changes in hospital admissions and complications of acute appendicitis during the COVID-19 pandemic: A systematic review and meta-analysis. Health Sci Rev (Oxf). 2022 Jun;3:100021. doi: 10.1016/j.hsr.2022.100021. 

6# I think many claims in the discussion need to be toned down

Reply: Thanks for your valuable comments. We have made some modification in the discussion to tone down the claims. We consider that comment#7 and #8 you put forward is what are needed to be tone down. 

7# Lines 320-322 not sure if just increase of CAT, SOD and GST-Px would mean ‘excellent oxidation resistance’. 

Reply: Thanks for your valuable comments. We have modified the sentence ‘produces excellent oxidation resistance ’ into ‘plays an important role in regulating the oxidative stress reaction’ to maintain the preciseness of discussion, which is shown in line 310 in the new version of manuscript.

8# Lines 399-401 How did you confirm “that acupuncture reduces the production of ROS and activates the antioxidant system in vivo” and how is it “evidence to prove the mechanism of acupuncture in treating multiple diseases”. This is merely a meta-analysis. 

Reply: Thanks for your valuable comments. We have removed previous statement you have pointed. The remodified statement is expressed as ‘The data from existing experimental studies suggest that acupuncture can regulate the oxidative stress status among multiple organs and tissues in animal models. However, more studies, especially clinical studies, are still needed to further explore and justify the oxidation resistance of acupuncture.’ This modification is shown in line 378-382 in the new version of manuscript.

9# Why is not the supplementary data cited in the text? How does one review it then.

Reply: Thanks for your comments. The supplementary data consists of two parts, one is PRISMA checklist to improve reporting quality of our study, the other is the sensitivity analysis. The objective of performing this analysis is to test each trial by meta after the consolidation and strengthen the credibility of the combined data. The results of this analysis does not affect the conclusions drawn from the research. Meanwhile, we describe this situation at the end of each indicator in the result section. 

Reviewer #3

Reviewer #3: The authors conducted a systematic review and meta-analysis of the published article assessing the effect of acupuncture on oxidative stress in animal models. The topic is relevant in the context of alternative medicine and may be of interest to the readers. However, following points need to be addressed. 

1# Literature search was undertaken from “inception to August 2021” (line# 28 and 117). What does inception mean?

Reply: Thanks for your comments. The word ‘incepton’ means the date when the database was initially established. Many published systematic reviews are expressed in this term within the search time range.We have cited some examples as follow: 

1. Li F, Wang L, Qin Y, Liu G. Combined Tai Chi and cognitive interventions for older adults with or without cognitive impairment: A meta-analysis and systematic review. Complement Ther Med. 2022, 67:102833. doi: 10.1016/j.ctim.2022.102833.

2. Shih CY, Gordon CJ, Chen TJ, Phuc NT, Tu MC, Tsai PS, Chiu HY. Comparative efficacy of nonpharmacological interventions on sleep quality in people who are critically ill: A systematic review and network meta-analysis. Int J Nurs Stud. 2022, 130:104220. doi: 10.1016/j.ijnurstu.2022.104220.

3. Cross AJ, Thomas D, Liang J, Abramson MJ, George J, Zairina E. Educational interventions for health professionals managing chronic obstructive pulmonary disease in primary care. Cochrane Database Syst Rev. 2022, 5:CD012652. doi: 10.1002/14651858.CD012652.pub2. 

2# The effect of acupuncture on other disease areas could have been discussed briefly in the introduction section as supporting information to discuss the relevance of this study.

Reply: Thanks for your valuable advice. We have added the effect of acupuncture on spinal cord injury in the introduction section as supporting information to discuss the relevance of this study, which is shown in line65 and line 83-85 in the new version of manuscript.

3# Please explain/elaborate what indicators (MDA, SOD, GSH-Px, CAT) are and discuss briefly how they are related to oxidative stress.

Reply: Thanks for your valuable comments. We have added related content about the explain of MDA, SOD, GSH-Px and CAT and their effect on oxidative stress, which is shown in line 300-306 in the new version of manuscript. 

4# In line# 15, what does “full text screening was then applied” mean?

Reply: Thanks for your comments. ”full text screening was then applied” means that the two researchers have finished reading the whole article carefully and extracted critical dataf or further analysis.

5# The font used in the manuscript is not reading friendly and the format of table 2 made it hard to follow.

Reply: Thanks for your valuable comments. The font of our manuscript has been changed into a more friendly format, and so was Table 2. 

6# Last line of conclusion (line# 405-408) seems irrelevant and does not fit to the context.

Reply: Thanks for your valuable comments. We have removed the corresponding statement.

7# Correction: “increased” in line# 36. Small letter in “criteria” in line# 29.

Reply: Thanks for your valuable comments. We have modified this in the new version of the manuscript.

---

## [Decision Letter · Decision Letter 1]

24 Jun 2022

The effect of acupuncture on oxidative stress: a systematic review and meta-analysis of animal models

PONE-D-22-01762R1

Dear Dr. Xia,

We’re pleased to inform you that your manuscript has been judged scientifically suitable for publication and will be formally accepted for publication once it meets all outstanding technical requirements.

Kind regards,

Ghulam Md Ashraf, Ph.D.

Academic Editor

PLOS ONE

Additional Editor Comments (optional):

Reviewers' comments:

Reviewer's Responses to Questions

**Comments to the Author**

1. If the authors have adequately addressed your comments raised in a previous round of review and you feel that this manuscript is now acceptable for publication, you may indicate that here to bypass the “Comments to the Author” section, enter your conflict of interest statement in the “Confidential to Editor” section, and submit your "Accept" recommendation.

Reviewer #2: All comments have been addressed

Reviewer #3: All comments have been addressed

2. Is the manuscript technically sound, and do the data support the conclusions?

Reviewer #2: Yes

Reviewer #3: Partly

3. Has the statistical analysis been performed appropriately and rigorously? 

Reviewer #2: I Don't Know

Reviewer #3: I Don't Know

4. Have the authors made all data underlying the findings in their manuscript fully available?

Reviewer #2: Yes

Reviewer #3: Yes

5. Is the manuscript presented in an intelligible fashion and written in standard English?

Reviewer #2: Yes

Reviewer #3: Yes

6. Review Comments to the Author

Reviewer #2: My comments have been addressed and I have no further comments. I leave it upto authors to annotate the figure legends. Although references provided showed similar legend description as described in this paper, I am more used to detailed legends.

Reviewer #3: (No Response)

7. PLOS authors have the option to publish the peer review history of their article (what does this mean?). If published, this will include your full peer review and any attached files.

Reviewer #2: No

Reviewer #3: No

---

## [Editor Report · Acceptance letter]

30 Jun 2022

PONE-D-22-01762R1 

The effect of acupuncture on oxidative stress: A systematic review and meta-analysis of animal models 

Dear Dr. Xia:

I'm pleased to inform you that your manuscript has been deemed suitable for publication in PLOS ONE. Congratulations! Your manuscript is now with our production department. 

Kind regards, 

on behalf of

Dr. Ghulam Md Ashraf 

Academic Editor

PLOS ONE